# Supporting and Retaining NHS England Staff with Long-Term Health Conditions—A Qualitative Study

**DOI:** 10.3390/healthcare13202573

**Published:** 2025-10-14

**Authors:** Jen Remnant, Moira Kelly, Laura Cowley, Sara Booth

**Affiliations:** 1Scottish Centre for Employment Research, University of Strathclyde, Glasgow G1 1XQ, UK; 2Centre for Primary Care and Public Health, Queen Mary University of London, London E1 4NS, UK; m.j.kelly@qmul.ac.uk; 3Patient Led Research Hub, Cambridge University Hospitals NHS Foundation Trust, Cambridge CB2 0QQ, UK; lbm28@medschl.cam.ac.uk (L.C.);

**Keywords:** long term health conditions, human resource management, NHS, employment

## Abstract

**Background**: NHS England has an ageing workforce. Approximately 30 percent of the NHS England workforce are aged 50 years and over, and the British Medical Association has argued that it is important that employers meet the needs of their ageing workforce and retain their skills and expertise. **Objective**: This sought to explore how NHS England Trusts support employees with fluctuating long-term health conditions, investigating systemic workforce challenges to providing adequate support and identifying opportunities for more inclusive and sustainable employment practices. **Methods**: Qualitative interviews were conducted with staff working in human resources, occupational health staff and clinical line managers involved in the support and management of staff with fluctuating long-term health conditions (n = 17). **Results**: The research found a misalignment between clinical managerial practices, human resource procedures, and the overarching NHS human resource policy framework, which was often seen as rigid and poorly suited to the fluctuating nature of some long-term conditions. These tensions were exacerbated by high staff turnover and limited organisational capacity. Nonetheless, instances of effective, person-centred support were also reported, typically occurring where cross-departmental collaboration and flexible, locally adapted approaches were in place. **Conclusions**: Findings suggest that targeted, flexible interventions for NHS employees with fluctuating long-term health conditions could enhance staff retention, reduce absenteeism, and promote more resilient workforce strategies. Identifying and scaling examples of good practice may be key to fostering a more inclusive and adaptive NHS employment model.

## 1. Introduction

NHS England directly employs 1.7 million people [1]. However, as is widely acknowledged, there are staff shortages across all job families in the service, including clinical staff [1]. It has been convincingly argued that alongside broader labour market, training and migration issues affecting recruitment, issues such as employee burnout and exhaustion are causing NHS employees to leave [2], that chronic workplace stress is contributing to negative working environments [3], and that absences caused by sick leave are increasing pressure on the remaining workforce [4].

NHS England struggles to fill vacant positions, leading to increased workloads for existent staff and potentially, compromised patient care [5]. Additionally, retention challenges exacerbate the ongoing staffing crisis as more healthcare workers are at risk of experiencing burnout [6], stress [7] and dissatisfaction with their working conditions [8]. These challenges have been particularly evident through recent industrial action across the devolved UK nation NHS services by professional associations [9] and trade unions representing NHS staff.

Unsurprisingly, the COVID-19 pandemic and subsequent lockdowns triggered an extraordinary intensification of the above issues, as well as impacting levels of absenteeism and presenteeism (attending work when not able to function adequately) across the NHS England workforce [10]. It placed unprecedented pressure on healthcare professionals and exacerbated existing workforce challenges. Frontline staff faced immense physical and emotional strain while responding to the pandemic, leading to heightened levels of stress, burnout and attrition [11,12].

NHS England workforce shortages disproportionately affect certain regions and specialties, deepening healthcare inequalities. Rural areas often struggle to attract and retain healthcare professionals, leading to limited access to services and sometimes poorer outcomes for patients [13]. Similarly, specialties such as mental health and primary care face significant workforce challenges, impacting timely access to essential services for vulnerable populations [14]. The context of this staffing crisis in NHS England is further compounded by the increasing demand for healthcare services driven by an ageing population and subsequent rising prevalence of long-term health conditions [15].

### Fluctuating and Long-Term Health Conditions (FLTCs) in the NHS Workforce

FLTCs are long-term conditions that are treatable but not curable, which entail fluctuating constitutional symptoms such as fatigue. Such conditions include systemic lupus erythematosus, rheumatoid arthritis and inflammatory bowel disease.

Corresponding with our ageing population, NHS England has an ageing workforce. Approximately 30 per cent of the NHS England workforce is aged 50 years and over [4] and the British Medical Association has argued that it is important employers retain skills and expertise by meeting the needs of their ageing workers [16]. One aspect of meeting these needs is recognising that the incidence of long-term ill-health and/or disability increases with age and that supporting an ageing workforce requires health and wellbeing-related resources [17,18,19]. Despite this, very little academic work has explicitly connected the support of ageing NHS workers with the provision of support for, or experiences of, disabled staff.

It is, admittedly, difficult to get a full understanding of disability and long-term ill-health in the NHS workforce due to the varied definitions of disability used across Trusts, localised and discretional negotiations of support, and low levels of disability disclosure [20]. However, research thus far paints a damning picture of the experiences of disabled NHS workers. There is evidence that even where low-cost resources were available, NHS Trusts have not provided appropriate workplace accommodations for disabled employees [21] and that some disabled NHS workers have been treated punitively when seeking support from their employers [22]. Many of the experiences reported by disabled NHS workers replicate broader research about the experiences of disabled workers; that support is dependent on managerial goodwill [23], informed by the manager-perceived deservingness of the individual employee [24,25], is subject to decision making by individuals with limited understanding of ill-health conditions and/or disability [26] and reliant on inadequate human resource policy [27]. There is also evidence that line managers and human resource staff find talking about disability difficult and are so concerned about saying the “wrong” thing that it inhibits open dialogue between themselves and disabled staff [28,29].

The ongoing mismanagement of disabled and long-term ill employees across sectors undermines the ability of organisations to reap the rewards of a diverse workforce. Though it has been recognised that there are strengths and difficulties related to supporting ag- and disability-diverse workforces [30], there is a substantial and growing body of work which evidences key benefits to supporting disabled workers across sectors. Findings suggest the effective management of disabled workers can result in improved workforce productivity [31,32], morale [33] and affective commitment (an employee’s perceived emotional attachment to their organisation) [34]—all of which could have clear benefits to NHS services across the devolved nations of the United Kingdom. Irrespective of the moral and business case for supporting disabled and/or ageing workers, the increased number of older workers is an inevitability that requires action, whether desired or not.

Enhancing accessibility and inclusion for disabled and long-term ill staff within NHS services holds the potential to yield positive impacts on organisational performance and staff wellbeing [35], as well as begin to address some of the issues related to high staff turnover. However, not enough is known about if, how, and when disabled staff and long-term ill staff receive appropriate support from their employers, who is providing and receiving adequate support, and/or whether good practice can be replicated between NHS Trusts. This paper presents a small study that addresses these issues and offers recommendations for practice and further research. The purpose of the study was to understand attitudes and knowledge of occupational health, human resources, and clinical managers of strategies available to them, and strategies they personally apply, to support and retain employees with FLTCs, and to understand conditions of good practice within NHS England Trusts.

## 2. Materials and Methods

The study design for this research was interview based, employing an abductive analytical approach, meaning that the research team drew on their prior knowledge of the topic to guide data collection but were open to new and unanticipated information from participants [36].

### 2.1. Patient and Public Involvement

This project was funded as a patient -ed project, as the original principal investigator (fourth author) was an NHS employee managing an FLTC and designed the project with colleagues in her local Purple Network. Other members of the research team had familial or personal experience of FLTCs and disability and are in academic or academic-adjacent roles. While we do not present data from individuals managing FLTCs, our focus on individuals who are directly responsible for the management and support of ill and disabled staff was directly informed by disabled and ill NHS staff. Four follow-up workshops were also conducted with the participating NHS Trust’s Purple Networks/disability forums to validate findings and inform analysis. Purple Networks/disability forums exist in several organisation types, including some NHS Trusts and are for staff with hidden or visible disabilities, physical, neuro-diverse or mental health conditions and allies to work together to promote inclusion. They include staff with fluctuating long-term health conditions. We did not collect participant demographic or occupational information or record the workshops due to the highly sensitive, personal and identifiable nature of the discussions.

### 2.2. Recruitment, Interviews and Workshops

Seventeen people were interviewed from across three NHS England Trusts, including 6 HR and 5 OH staff and 6 NHS clinical line managers. The labels given to participants in the text denote the order they were due to be interviewed in (Int#), the Trust they were employed by (T#) and their role (CM/OH/HR). One participant withdrew. The participant was not asked for a reason for their withdrawal. Information on which Trusts the participants were recruited from is summarised in Table 1. Participants were observed to be predominantly women (n = 14), which reflects the disproportionate overrepresentation of women in NHS workforces. No further demographic information was requested, partially in recognition of the time constraints on participants and the effort to focus the time on responding to the research questions, and partially to avoid collecting unnecessary data. As such, we do not share other demographic data other than to explain that the participants were observed to be predominately women. We have also elected not to provide the specific times and dates of the interviews, only the time frame within which they were completed, as is standard for peer review.

The benefit of interviewing HR and OH clinical staff is that they were able to draw on their experiences of working with multiple managers across their respective Trusts. Further participant information is available in Table 2.

Participating Trusts were selected to ensure as much representativeness as is possible with qualitative research: they vary in size and service provision (including specialist mental health services) and provide healthcare to rural, semi-rural and urban populations. This information is summarised in Table 3. The research team opted for a broad variety across the selected NHS Trusts for the study to enable the exploration of commonalities in experience. The interviewer (second author) is an experienced qualitative researcher with knowledge of the NHS.

Participants were recruited via contact with Trust HR departments or via snowball sampling from previous participants. The project received favourable ethical opinion from the University of Strathclyde (May 2022) and was exempt from the NHS research ethics committee review as it was categorised as a service evaluation.

Participants gave written informed consent and were interviewed using either Microsoft Teams or Zoom while in their places of work. All interviews were digitally audio recorded with permission, transcribed by a trusted service, and anonymised. The interviews were semi-structured and focused on participant experiences of providing support to employees with FLTCs. This approach was utilised to fully address the research objectives of the project, while also remaining flexible to unanticipated additions or comments from participants [37]. Data were collected over a period of 7 months, from September 2021 to March 2022. Each interview lasted approximately 60 min. The interviewer (second author) is a female medical sociologist (PhD) who is an experienced qualitative researcher. She had no previous contact with participants. Although interviews were conducted during the height of the COVID-19 pandemic, this paper focuses on data relating to the management of FLTCs and includes participant reflection on supporting employees before and during the pandemic. Participant experiences specific to COVID-19, as captured in interviews, will be explored in a further publication. Recruitment activities ceased when the team agreed that only extant themes were emerging from interviews. This is often referred to as data saturation, though authors recognise that this is a contested concept [38]. Ending recruitment was also informed by practical limitations, including difficulties NHS staff had in finding the time to participate and the overall scheduling of the project.

Follow-up workshops, two online and two in-person, were advertised to members of either the Purple Network or the disability forum at each of the three Trusts. The first three authors facilitated the workshops. Attendees were presented with findings from the project and invited to provide feedback and reflection based on their own lived experience. To encourage open and transparent dialogue, attendee details (demographic or occupational) were not collected or stored beyond the purpose of organising refreshments and room size for the events. At the beginning of each workshop, the research team sought permission from attendees to take thorough notes to inform research team data coding discussions and the development of the study recommendations. These were the only fieldnotes taken during this study.

### 2.3. Analysis

Data from this study were treated as confidential and kept secure in compliance with the relevant UK General Data Protection Regulation (GDPR), the Data Protection Act, 2018. The anonymised transcripts were uploaded to NVivo 10 exclusively for data management, not for analysis, which was conducted by the research team through constant comparison.

The development of analytical codes and themes for this paper was informed by a “practical compromise of induction and deduction”, realistically capturing the process by which the subsequent theorising occurred [39] p. 79. This included initial coding alongside conducting interviews by the interviewer (second author) and full team discussions of the data during fieldwork. Once all transcriptions were completed, the full team discussed the emergent themes identified by the researcher, which provided a framework for the first author who coded the data in NVivo. This resulted in the identification of [N] themes, inclusive of [N1] subthemes. These themes were discussed and reviewed by the full team to refine and get consensus. The developing themes were also discussed in the workshops with Disability/Purple network members at the participating trusts for validation. We do not present workshop data in this paper but can confirm that workshop participants reported that our findings aligned with their experiences of management and HR engagement. Overall, analysis identified over twenty themes, three of which are presented in this paper: (1) the conflict between clinical managers and HR support for employees with FLTCs (2) the (perceived) inflexibility of NHS Human Resources policy and practice and (3) uncontrolled turnover and change.

## 3. Results

The section presents interview findings in the three discrete themes listed above before discussing examples of good practice identified in the data. The clinical manager—HR divide.

Throughout data collection, it was apparent that there is extensive variation in how people with FLTCs are managed. This variation is largely due to the high level of managerial discretion in decision making about employee support, where clinical managers are often individually responsible for identifying and accessing relevant information to support their employees. The interviewees explained how, in their experience, managers commonly felt ill-equipped to fulfil their responsibilities to employees with FLTCs. Participants in HR roles highlighted how manager concerns covered all aspects of supporting their employee, including some around employee disclosure and feeling unsure about appropriate terminology for discussing ill-health and disability with employees, which was described as “a real minefield” (Int10T1HR4), through to the operational difficulties associated with managing someone with an FLTC, as explained by a member of occupational health staff here:
*“I think again from a manager’s point of view I think that can feel quite hard to manage because it’s unpredictable… trying to run a service or a department not knowing whether someone’s going to be able to be on top form on any given day is probably quite hard…”*(Int14T2OH)

Clinical manager participants specifically identified key difficulties when trying to support employees at the early stages of their condition. Several participants commented on the difficulties in supporting someone who had not disclosed a condition to them but appeared to be symptomatic and/or taking sick leave. Managers described how in these situations they were not sure whether it was appropriate to raise it with the employee.

Participants also acknowledged how managerial responsibility extended beyond employees with FLTCs and included managing the concerns of the remaining team members. One clinical manager outlined concerns they had about being perceived to be treating one employee differently from others because of their health status:
*“It’s also around the perception of others. ‘They are always off sick so why aren’t you doing something to them’ ‘why are they special’ ‘why are they allowed off sick so often’. What it will look like is that they are off sick quite often but nobody is doing anything about it.”*(Int8T3CM)

What is interesting about this quote is that while the manager does recognise that it is their responsibility to manage employee relations between their direct reports, they do not recognise that these other team members might require additional information or have training needs relating to workforce diversity. Here, the clinical manager situates providing support to a staff member experiencing ill-health as a cause of potential disharmony, rather than an issue of communication and learning for the whole team. This was a recurring sentiment throughout the data and was validated at our workshops.

OH and HR participants reported that clinical managers often expressed concern to them about their lack of condition-specific knowledge relating to the symptoms and/or diagnoses their employees had disclosed to them. They commented on how managers wanted more information on specific symptoms and illness trajectories. However, OH and HR staff argued that this information was not necessarily relevant to providing adequate support:
*“…the NHS, it sometimes gets a little bit complicated… managers feeling they need more information about the health condition [of their employee], how it’s managed etc., where actually that’s not relevant, they just need to know what adjustments are needed.”*(Int14T2OH)

Rather than focus on HR processes, clinical managers typically seek to support staff with FLTCs by understanding the specifics of the healthcare condition and sometimes reserve judgement on support until they receive a specialist referral and advice. Focusing on the condition conflicts with the HR and OH approach to focus on the adaptive support an employee needs to be able to perform in their role. While focusing on the “problem” may cause delays in providing support given the extensive waits associated with some FLTC diagnoses [40,41], it is clear a balance between individualised support and expediency is required.

This discrepancy was further amplified by participants who felt that managers did not necessarily consider HR processes as part of their managerial responsibility:
*“… I don’t think we train managers well enough to know there’s an expectation that they know how to manage a disciplinary, sickness absence, capabilities”*(Int14T2OH)

The data is indicative of a mismatch between what clinical managers feel they should offer to employees with FLTCs, the usefulness of existing HR processes, and the HR processes they believe they are responsible for.

### 3.1. Inflexibility of HR Policies and Processes

In addition to the misalignment between clinical managers, OH and HR staff, participants also identified issues with both formal HR policies and informal HR processes within NHS Trusts. Clinical managers were regularly seen to experience difficulties when trying to effectively operationalise these procedures to support employees with FLTCs.

Clinical managers explained how they felt that engaging with HR policies was largely performative, to evidence that they had followed procedure rather than something that they felt was beneficial to their employees:
*“…you need to be seen to be doing the right thing and there is a process to follow because, at the end of it, if things become—don’t work out or there’s a lot of angst and anger then when you sit down with HR you can say, “I have followed this, done the right things up until now but we aren’t getting to where we need to be.”*(Int11T3CM)

Rather than viewing HR policies and procedures as enabling or helpful, the above manager categorises them as an inconvenient but necessary protection from complaint if their employee is upset at the support and/or management they have received. The same participant went on to say that the only value to engaging with the policies and procedures was that HR colleagues knew ‘*…you’ve done what you’re supposed to do*’ irrespective of the outcome for the employee with an FLTC.

Managers explained how they felt HR policies lacked the flexibility they needed to support employees with FLTCs. They described being directed toward the policies by HR staff rather than receiving the bespoke, employee-specific guidance and support they were seeking:
*“…you’d go to your line manager or HR and say, right I’ve got this member of staff that needs, they’ve got this… and often [they] come back and go ‘oh there’s a policy, you should read that and follow this’ and yes they are great for a basic [introduction] but they are not about that person in particular.”*(Int12T3CM)

This implies that clinical managers are not aware of their own discretion regarding the application of HR responsibilities or may lack experience and confidence in decision making. Instead, expect individualised advice specific to employee needs. It also suggests that this gap between the support they would like and the support that they get from HR contributes to this sense of their engagement with HR policy and procedure being performative rather than operational. The gap is further exacerbated by the differences in approach as discussed above. It is of note that managers who were more experienced were more confident in supporting employees, for example, in identifying flexible working options.

This issue was evident across the interview data, where managers gave examples of times where they felt discretion was appropriate but believed it contradicted with what they were supposed to do according to Trust policy:
*“…if I had somebody in front of me that had recently had a family bereavement and couldn’t cope at work, I would talk to them in a totally different way to somebody who you know from looking at the rosters is taking regular sick days… It’s two totally different conversations but, in theory, they’re under the same policy.”*(Int11T3CM)

This quote is interesting because, while drawing attention to the perceived inflexibility of the policy, the manager also alludes to how they would treat employees differently depending on their personal circumstances.

Further to having concerns about Trust HR policies, many participants felt that resources available to support managers with employees with FLTCs were misused or misunderstood. For example, participants described redeployment as a “last resort really” (Int12T3CM3) that was viewed negatively and not well managed. HR and OH participants also explained how OH services could be viewed as a punishment rather than support by managers:
*“…I think people often view occupational health in a slightly punitive way. So, if you don’t start attending work, I’m going to send you to occupational health… a referral to occupational health happens when somebody’s sickness absence has been very high for any reason… it’s a fairly blunt tool.”*(Int13T2OH)

Participants also described how some clinical managers ignored OH advice, understanding it to be advisory, rather than binding.

### 3.2. Turnover and Stress in the NHS

All participants reflected on how NHS staff have been managing higher workloads and increased stress over the last decade, which at the time of the interviews was compounded by the additional pressures caused by the COVID-19 pandemic. They explained how this meant clinical managers had less time *“to do their people management stuff… so that takes away their time from being able to sit down with someone and say, ‘you’ve come back from sick leave and how are you feeling—are you fit to be back from work? Let’s talk about”* (Int17HR).

Additionally, participants reported that clinical managers were themselves experiencing burnout and stress, sometimes resulting in absence, even prior to COVID-19. It was clear that managers felt that the pressure on their services and the NHS more generally meant that they did not have the necessary resources, physical or emotional, to provide the inclusive, altered environments their employees with FLTCs needed. This was reflected in the testimonies of HR and OH participants, who outlined the communication strategies they undertook to encourage managers with workplace accommodations. This often meant emphasising how supporting small changes can lead to substantial gains:
*“…I do explain to them, well, if you [the manager] can make some short-term changes at the moment, that could make a very big, long-term difference [for the individual]”*(Int13T2OH2)

Participants also reflected on how they felt unable to change someone’s material circumstances at work, which was identified by many as being a cause of some FLTCs related to poor mental health. An OH participant explained:
*“If somebody is off for a month because they felt depressed and they get better and then go back into exactly the same situation they were in before, then the same thing is going to happen again”*(Int16T1OH)

All participants situated the provision of workplace accommodations as something that increased managerial workload, rather than something that would support staff to work at their best and ultimately reduce the strain on managers. This does not represent individual poor practice but instead draws attention to how organisational expectations for clinical managers to provide individualised support for employees with FLTCs, and disabled staff more generally, inhibits the provision of adequate support as it becomes another tick-box item for managers already at capacity.

### 3.3. What Does Good Practice Look Like?

An important objective of this research was to explore what good practice looked like in terms of supporting colleagues with FLTCs and what the conditions were for providing good support. Unsurprisingly, the good practice identified related to managerial actions that involved working collectively and engaging in knowledge-sharing activities. In basic terms this meant asking for guidance from more experienced colleagues and discussing options and contacts.

Collective actions were discussed by participants with examples of when teams “pulled together” to provide care and support for their colleagues. Sometimes this involved a redistribution of tasks to better accommodate the strengths and limitations of different team members, while others included support beyond the remit of the workplace:
*“…three or four of us went round to the house for an afternoon and we blitzed [cleaned] it from top to bottom… those little personal things meant more than anything else”*(Int11T3CM)

Though this example extends beyond role expectations, it is indicative of a team thought process at odds with the organisational norms highlighted in the previous sections. It is the assumption of collective support and responsibility that is of analytical interest.

Further to this, it was interesting that in these instances of perceived success in providing appropriate support, managers did not discuss policies, procedures or formal guidance. They also did not discuss the varied Trust resources available. Where they felt that they offered useful, good practice, it related to human interaction, rather than technology or resources. This indicates that opportunities to discuss a collaborative approach to resolving workforce needs could be of benefit, particularly between more and less experienced managers. However, further research is needed to identify whether the perceived success of managers aligns with what employees with FLTCs consider adequate support and whether this bears out in employee turnover.

## 4. Discussion

This study underscores the complexities surrounding the effective management of NHS staff with FLTCs. As highlighted in previous literature, the NHS staffing crisis is exacerbated by factors such as burnout and increased workload, which are compounded by the ageing workforce, the rising demand for healthcare services [1,15] and, though not discussed at length in this paper, the COVID-19 pandemic. We also know from research outside the UK that poor managerial leadership may increase sickness absence among the health and social care workforce more generally [42]. Our research elucidates the critical role that managerial practices play in the retention of employees with FLTCs, revealing a landscape marked by variability in support and disconnect between HR policies and the needs of individual employees, which may evolve over time.

This research identified three key themes that impact the support provided to NHS employees with FLTCs. The first is the individualised, discretional approach to managing employees with FLTCs in the NHS. The second is the inflexibility, both real and perceived, of NHS HR policies and procedures. The third relates to the state of flux most participants viewed the NHS to be in regarding staff turnover and stress. Though presented separately, it is important to recognise that these themes are interrelated and need to be reflected on in unison. An important inhibitor for clinical managers trying to provide adequate support to their employees with FLTCs is the contradiction between the level of managerial responsibility and discretion in providing support and the perceived inflexibility of HR policies and processes. In addition, uncertainty in the early stages of illness and fluctuating symptoms impede the identification of appropriate support.

A feature of the research is the evident divide between clinical managers and HR and OH staff in their approach to supporting employees with FLTCs. Clinical managers tend to individualise support based on the specifics of an employee’s condition. In this respect, our data contrasted with existing research that argues that employee support is often subject to decision making by individuals with limited understanding of ill-health conditions and/or disability [26]. In the data we present, it appears that for some clinical managers, their understanding, or desired understanding, regarding the specifics of their employees’ health conditions was perhaps too precise. Our data suggest that the focus on condition trajectory fails to accommodate employee support needs (the focus of HR participants) or employee experiences of exclusion or ableism.

This study extends work that has shown that managerial support is dependent on personal goodwill [23,24]. It also highlights how it is dependent on perceived managerial resources, which is a key issue within an individualised model of support provision in (an) organisation(s) in crisis [1,3]. In this context particularly, clinical managers may feel overwhelmed when responding to the needs of their employees and seek safety and assurance through organisational policy. However, despite being collective in terms of their coverage, such policies do not support a collective organisational responsibility for supporting employees. Instead, replicating other organisations [27], HR policies and procedures accessed by NHS managers are perceived to be “one size fits all” and lack appropriate structure to accommodate the fluctuating nature of many healthcare conditions and challenges associated with symptoms, navigating diagnosis, prognosis and treatment regimens alongside work commitments [41]. These challenges extend beyond the affected staff member to their team, manager, and wider colleagues.

While NHS HR policies intend to guide managers in supporting employees with FLTCs, our findings reveal that these policies are often perceived as inflexible and performative. Managers reported that HR policies did not account for the nuances of individual circumstances, leading to frustration and a reliance on personal discretion to navigate employee support. This dissonance highlights a critical gap in the implementation of HR practices within the NHS, where adherence to policy can overshadow the actual needs of employees, which may be fluctuating and changeable.

The challenges identified in our study are not isolated; they contribute to broader issues of turnover and employee wellbeing within the NHS. The stress of managing a team under conditions of uncertainty, compounded by the pressure to maintain service delivery, leaves little room for the compassionate management of health-related absences. Our participants highlighted that many managers were already stretched thin, leaving them unable to prioritise meaningful conversations with staff returning from sick leave.

This lack of engagement not only affects individual employee outcomes but also has the potential to impact team morale and retention overall. Creating a culture of open dialogue about health conditions and fostering an inclusive environment could lead to better outcomes for both employees and the organisation. Opportunities for knowledge sharing between managers and more experienced staff—including those with lived experience of managing FLTCs—and generating opportunities to discuss health and wellbeing in both formal and informal settings may improve workplace dynamics and reduce the stigma surrounding ill-health. Disabled and ill healthcare workers can advocate for necessary accommodations that not only improve their quality of practice but also foster a more inclusive healthcare environment for patients requiring similar supports, such as accessible facilities and understanding regarding their health conditions [43].

## 5. Conclusions

This paper contributes to the literature by addressing the underexplored experiences and perspectives of those responsible for managing NHS staff with FLTCs, highlighting how the unpredictability of such conditions exposes tensions between individualised managerial discretion and the perceived inflexibility of HR policies. Unlike prior research that frames support as limited by managers’ lack of understanding, this study reveals that focus on condition trajectories can hinder effective support. It identifies a divide between clinical managers and HR/OH staff, showing how standardised, “one size fits all” policies fail to accommodate the fluctuating needs of employees, leaving responsibility to overstretched managers. By linking these issues to wider problems of turnover, morale, and workforce sustainability, the study makes empirical, theoretical, and practical contributions, underscoring the need for more flexible HR frameworks, shared organisational responsibility, and inclusive practices to better support employees with FLTCs.

## 6. Recommendations for Practice and Future Research

For practice recommendations we are cognisant of the findings relating to managerial pressures, and as such, have focused on recommendations that can be enacted in contexts of limited budget and time resources.

Add agenda points to all business-function meetings on how to pre-emptively manage health diversity in the workforce to all workforce meetings. This suggestion is to utilise existing team meetings, led by clinical and non-clinical managers who do not have specific EDI roles, to facilitate discussions on the ongoing management of health-diverse teams that are not siloed as issues exclusively related to Equality, Diversity and Inclusion, as is commonly the case in large organisations [27].

Harness the HRM policy review process, led by senior HR advisors, to develop clear, summarised guidance materials about managerial HR responsibilities. The data we present imply a mismatch between what managers think HRM policies and procedures are for and what HR staff think they are for. There is also scope to provide more practical training for clinical managers about these responsibilities rather than more generic equality, diversity and inclusion training. For example, skills in communicating collaboratively with staff with FLTCs to explore options for support.

Develop more centralised processes for recording workplace accommodation requests. Though this is not necessarily practicable Trust-wide, compiling and sharing existing workplace accommodations at specific sites will generate useful data for understanding employee needs collectively, which could impact workplace design and the wider organisation of work in some services. The impact of these recommendations could be measured through existing workforce surveys and reports.

## 7. Study Limitations

There are key limitations to this study, including its small sample and limited geographical coverage, which only included representatives from NHS England Trusts, though the findings do support wider research regarding the management of healthcare workforces outside the UK, so remain relevant [42,43,44,45] Additionally, though employees with FLTCs co-designed the research project and validated the findings, they are not directly represented in the data. We did not collect demographic information from participants, which has meant we were not able to include such details in our analysis, and lastly, data were collected between September 2021 and March 2022, meaning that more up-to-date data are required to reflect the current challenges faced by NHS staff.

Our study has explored issues identified in earlier research with employees with FLTCs [22]. We recommend that our study findings can inform further research exploring the experiences of employees with FLTCs in more depth, focusing on their perspectives on the support they receive and the barriers they encounter in combination with data from other workplace stakeholders. Investigating differences in support across various NHS Trusts and regions may also provide valuable insights into best practices that can be shared and implemented widely, and this is likely to require mixed-method research which uses workforce data such as responses to NHS iMatter surveys in Scotland.

## Figures and Tables

**Table 1 healthcare-13-02573-t001:** Participant information.

	Trust 1	Trust 2	Trust 3	Total
Clinical Managers	3	0	3	6
Human Resources	4	0	2	6
Occupational Health Clinicians	1	4	0	5
Total	8	4	5	17

**Table 2 healthcare-13-02573-t002:** Participant experience working in NHS services and managerial responsibilities at the time of interview.

Participant	Years in NHS	Responsibilities
Int1T2OH	37	Manages team of 16
Int2T1HR	4	Directly manages team of 1, HR responsibilities for over 2800 staff
In3T1HR	25	Directly manages team of 5, indirect management of a further 5
Int4T1HR	10	No direct management responsibilities reported
Int5T1HR	40	No direct management responsibilities reported
Int6T3HR	4	No direct management responsibilities reported
Int7T1CM	28	Direct manages team of 26
Int8T3CM	41	Directly manages team of 9, indirect management responsibilities for over 2000 staff
Int9T1CM	20	Directly manages 1, indirect management for 30
Int10T1HR	9	No direct management responsibilities reported
Int11T3CM	44	Direct management responsibilities reported, changeable number of staff
Int12T3CM	30	Direct manages team of 8, indirect management of 50
Int13T2OH	22	No direct management responsibilities reported
Int14T2OH	23	No direct management responsibilities reported
Int15T2OH	<1	No direct management responsibilities reported
Int16T1OH	35	No direct management responsibilities reported
Int17T1CM	32	Shared management of 180 staff

**Table 3 healthcare-13-02573-t003:** Approximate Trust workforce numbers during data collection.

Trust 1	Approximately 12,000 staff
Trust 2	Specialist Mental Health Trust, over 1500 staff
Trust 3	Approximately 10,000 staff

## Data Availability

The data presented in this study are available on request from the corresponding author. The data are not publicly available due to privacy or ethical restrictions.

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
