# Peer review of "Supporting and Retaining NHS England Staff with Long-Term Health Conditions—A Qualitative Study"

_healthcare, 2025, doi:10.3390/healthcare13202573_

Round 1

Reviewer 1 Report

Comments and Suggestions for Authors

Dear Authors,

Thank you for the opportunity to review your manuscript. I appreciate the attention you bring to the important and timely topic of supporting NHS staff with fluctuating long-term health conditions (FLTCs). The paper makes a valuable contribution by shedding light on the tensions between HR processes, occupational health, and line management, and by offering pragmatic recommendations for practice.

That said, there are several areas where the manuscript would benefit from significant revision before it can be considered for publication. My comments are intended to help you strengthen the clarity, methodological transparency, and overall contribution of your work.

  1. Sample consistency. There is a discrepancy between the reported number of participants (17) and the role counts provided (6 HR, 6 OH, 6 line managers = 18). In addition, some quotations are attributed to “Int18,” suggesting more than 17 interviews. Please reconcile the participant numbers and interview identifiers for consistency.
  2. Abstract and keywords. The abstract currently contains a placeholder for keywords, and the final keyword list is presented separately. Please remove the placeholder and finalize a single, journal-compliant set of keywords. Consider whether a structured abstract might make your contribution clearer.
  3. Scope of claims. The study is based on NHS England data, but several narrative passages generalize to the UK as a whole. Please align claims with your sampling frame and clarify when broader UK context is background rather than evidence-based conclusion.
  4. Methodological transparency. While you state that an abductive approach and NVivo were used, the paper would be strengthened by more detail on:
    • How codes and themes were developed, refined, and validated.

    • The number of coders and processes for resolving differences.

    • Consideration of data sufficiency or saturation.

    • Reflexivity and positionality of the researchers.

    • How the Purple Network workshops influenced theme development.
      A participant characteristics table (roles, Trust type, region, experience) and a themes-by-quote table would also enhance transparency.

  5. Ethics and data protection
    Please update references to the Data Protection Act 1998. The relevant legal framework is UK GDPR and the Data Protection Act 2018. Ensure consistency in how approvals and exemptions are described across sections.

  6. COVID-19 context
    Interviews were conducted during the pandemic (Sept 2021–Mar 2022), but COVID-specific content is to be reported elsewhere. It would help readers to understand how pandemic conditions may have influenced the findings and whether these tensions persist outside of crisis contexts.

  7. Novelty and contribution
    Please strengthen the discussion by articulating more clearly what this study adds to existing literature on disability, reasonable adjustments, and line manager discretion in healthcare. A concise “value-added” paragraph would help.

  8. Recommendations
    Your practice recommendations are valuable. To increase their utility, consider indicating who within Trusts might be responsible for implementing each, what minimum resources are required, and how impact might be measured.

I encourage you to revise the manuscript in line with the above feedback. With stronger methodological detail, consistency, and sharper positioning, this study has potential to make a meaningful contribution to the literature on disability and workforce retention in healthcare.

Author Response

Dear colleague, thank you for the opportunity to revise our manuscript. Below we have outlined how we have responded to each of the suggested changes.

Reviewer 1

  1. Sample consistency. There is a discrepancy between the reported number of participants (17) and the role counts provided (6 HR, 6 OH, 6 line managers = 18). In addition, some quotations are attributed to “Int18,” suggesting more than 17 interviews. Please reconcile the participant numbers and interview identifiers for consistency.

Thank you for this observation. We have corrected the error and clarified that there are 17 participants. Participants were provided with an ID at first contact, and some withdrew, meaning the numbers do not align with overall participation, but with initial expressions of interest. We have altered our labels so that labelling is no longer confusing. We have clarified this in text, please see from line 143 in the tracked changes document.

  1. Abstract and keywords. The abstract currently contains a placeholder for keywords, and the final keyword list is presented separately. Please remove the placeholder and finalize a single, journal-compliant set of keywords. Consider whether a structured abstract might make your contribution clearer.

We have restructured the abstract as per your above suggested and corrected our keywords.

  1. Scope of claims. The study is based on NHS England data, but several narrative passages generalize to the UK as a whole. Please align claims with your sampling frame and clarify when broader UK context is background rather than evidence-based conclusion.

We have checked our sources/references and referred to NHS England throughout where we are using NHS England statistics, and more clearly articulating the scope of our claims in all sections of the paper.

  1. Methodological transparency. While you state that an abductive approach and NVivo were used, the paper would be strengthened by more detail on:
    1. How codes and themes were developed, refined, and validated.
    2. The number of coders and processes for resolving differences.
    3. Consideration of data sufficiency or saturation.
    4. Reflexivity and positionality of the researchers.
    5. How the Purple Network workshops influenced theme development.
      A participant characteristics table (roles, Trust type, region, experience) and a themes-by-quote table would also enhance transparency.

Thank you for these suggestions. We have added additional detail on how themes were developed and refined. Please see from line 206 in the tracked changes document. This section also confirms that while authors one and two were primarily responsible for coding, the full authorship team was involved in the refinement and development of project themes. We also confirm that these themes were validated with NHS staff members of Purple/Disability networks.

We also explain that we ceased recruitment activities due to a combination of team agreement that only extant themes were emerging in later interviews, which is sometimes referred to as data saturation, and practical limitations relating to project timelines.

We have added tables that summarise the Trusts participants were sampled from, summarised Trust information and pertinent participant occupational characteristics. You rightly identify that we do not provide a lot of detail about the workshops with the purple/disability networks. We opted not to collect participant demographic or occupational information due to the highly sensitive, confidential and identifiable nature of the discussions in these workshops. Please see line 141 in the tracked changes document.

We acknowledge your suggestion of a themes-by-quote table but feel that our revised explanation of our theme development adds enough transparency without this addition.

We have also added more information about the research team for additional positionality context.

  1. Ethics and data protection
    Please update references to the Data Protection Act 1998. The relevant legal framework is UK GDPR and the Data Protection Act 2018. Ensure consistency in how approvals and exemptions are described across sections.

Thank you for this observation, we have corrected the errors in text. Please see from line 208 in the tracked changes document.

  1. COVID-19 context
    Interviews were conducted during the pandemic (Sept 2021–Mar 2022), but COVID-specific content is to be reported elsewhere. It would help readers to understand how pandemic conditions may have influenced the findings and whether these tensions persist outside of crisis contexts.

Thank you for this suggestion. We explain in the paper that though the interviews were conducted during the COVID-19 pandemic, many of our interview questions related to how staff with FLTCs were supported and managed before and during the pandemic. The themes presented in this data were consistent in testimony relating to before and during the pandemic, and therefore, we believe not specific to this time frame. However, we have added additional detail to section 3.2, regarding turnover and stress to emphasise that these issues were compounded during the pandemic. 

  1. Novelty and contribution
    Please strengthen the discussion by articulating more clearly what this study adds to existing literature on disability, reasonable adjustments, and line manager discretion in healthcare. A concise “value-added” paragraph would help.

Thank you for this. We have incorporated this into a conclusions section. Please see section 5, line 471 in the tracked changes document.

  1. Recommendations
    Your practice recommendations are valuable. To increase their utility, consider indicating who within Trusts might be responsible for implementing each, what minimum resources are required, and how impact might be measured.

We have developed our recommendations to include further information about potential key agents, the resources required to implement them and the mechanisms for their measurement.

Reviewer 2 Report

Comments and Suggestions for Authors

Dear authors,

The manuscript addresses an important topic and is overall well written, with a clear structure, rigorous qualitative approach, and practical recommendations that will be of interest to both scholars and NHS practitioners. The abductive design, the use of semi-structured interviews across different Trusts, and the inclusion of validation workshops strengthen the credibility of the findings. Quotations are well integrated, and the thematic presentation is coherent.

However, there are some points requiring further attention to strengthen transparency and adherence to qualitative reporting standards (COREQ):

  1. Abstract: avoid abbreviations (NHS, HR, OH, FLTCs) and remove references. Abstracts should remain self-contained, without citations. The first use o abbreviations should be reserved for the main body, starting in the introduction.

  2. Researcher reflexivity (COREQ Domain 1): provide more detail on the interviewer’s characteristics (e.g., gender, credentials, position, prior relationship with participants, and any potential biases or motivations). This enhances transparency and credibility.

  3. Participant selection and non-participation: while recruitment is described, please indicate whether any individuals declined participation or withdrew, and the reasons if known.

  4. Data saturation: clarify whether saturation was reached, how it was assessed, or explain why it was not addressed.

  5. Field notes: state whether field notes were made during/after interviews.

  6. Data analysis: specify the number of coders involved and whether an analytic coding tree or framework was developed.

  7. Limitations: while you acknowledge sample size and lack of diversity, the absence of direct perspectives from employees with FLTCs themselves should be explicitly highlighted as a limitation.

  8. Literature base: the discussion would be further enriched by including additional international, peer-reviewed evidence on managing chronic illness and disability in healthcare workforces, to extend generalisability beyond the UK.

These revisions are relatively minor and do not detract from the overall quality of the paper. With these clarifications, the manuscript will be methodologically stronger and fully aligned with COREQ standards.

Kind regards

Author Response

Reviewer 2

  1. Abstract: avoid abbreviations (NHS, HR, OH, FLTCs) and remove references. Abstracts should remain self-contained, without citations. The first use of abbreviations should be reserved for the main body, starting in the introduction.

We have made this change as suggested.

  1. Researcher reflexivity (COREQ Domain 1): provide more detail on the interviewer’s characteristics (e.g., gender, credentials, position, prior relationship with participants, and any potential biases or motivations). This enhances transparency and credibility.

We have added more information about the full research team, as well as specific information about the researcher. Please see line 129-135 and lines 184-186.

  1. Participant selection and non-participation: while recruitment is described, please indicate whether any individuals declined participation or withdrew, and the reasons if known.

We have explained that one participant withdrew, and no reason was given or asked for. Please see line 148 of the tracked changes document.

  1. Data saturation: clarify whether saturation was reached, how it was assessed, or explain why it was not addressed.

We have added explanation that we ceased recruitment activities due to a combination of team agreement that only extant themes were emerging in later interviews, which is sometimes referred to as data saturation, and practical limitations relating to project timelines. Please see lines 140-143 in the tracked changes document.

  1. Field notes: state whether field notes were made during/after interviews.

Fieldnotes were only taken during the workshops, to support research team coding discussions. This is clarified on section 2.2.

  1. Data analysis: specify the number of coders involved and whether an analytic coding tree or framework was developed.

Thank you. We have provided further detail on our coding in section 2.3. This includes a description of our coding process, including development, refinement and validation.

  1. Limitations: while you acknowledge sample size and lack of diversity, the absence of direct perspectives from employees with FLTCs themselves should be explicitly highlighted as a limitation.

Thank you for this suggestion. We have added a study limitations section (section 7) which addresses all of these points. We have also added additional detail on the patient-led nature of this study (section 2.1) where we highlight that though we had no participants with FLTCs, the design of the project and focus on the perspective of those with workplace decision-making powers was developed in partnership with NHS staff with FLTCs, and so their voice is present.

  1. Literature base: the discussion would be further enriched by including additional international, peer-reviewed evidence on managing chronic illness and disability in healthcare workforces, to extend generalisability beyond the UK.

There is a notable absence of research globally about the management of LTCs in healthcare workforces in the UK and elsewhere in the world. We have added some reflection on this to the discussion and limitations sections (sections 4 and 6 respectively).

Reviewer 3 Report

Comments and Suggestions for Authors

You can find Review in attached file

Author Response

Reviewer 3

  1. Shortcomings which cannot be addressed:
    1. The small sample (17 participants) is insufficient for deep generalization
    2. Employees with FLTCs were not involved in the interviews - only managers, HR and OH. As a result, the conclusion about "good practice" is based on the opinion of managers, and not from service recipients (Employees with FLTCs)
    3. Limited geography - only three trusts in England. Perhaps the state of HR policy in other regions of the UK is different
    4. The time of data collection includes the COVID-19 pandemic. It is possible that the stress load increased during this period.

Thank you for these observations. We have added clarifications to ensure readers are aware that this research funding was patient led, meaning that the principal investigator (fourth author) had a FLTC, and co-designed the research with other NHS staff with FLTCs. While none of the participants identified as people managing FLTCs, the voice of people with FLTCs is implicit in the study design and focus on the individuals who make decisions with material implications for the management of ill and disabled staff. Please see line from line 143 in the tracked changes document.

  1. Shortcomings that can be addressed:
    1. Lack of information on age, gender, ethnicity, level of experience. For example, it is possible that the opinions of young and old managers differ.
    2. Information about the workshops is extremely limited. There is no data on the composition of the participants. The article does not describe the specific process of synthesis (for example, how the notes were coded, how many themes were identified, or how discrepancies were resolved). This makes the conclusions subjective.

Thank you for these observations. We have provided relevant additional information about the length of experience of each of the participants in table 2 (please see line 154 onward). In the text we clarify that we did not ask participants for irrelevant demographic information as we were not analysing the data based on gender, ethnicity or age. We consider it good ethical practice to only ask for information that is pertinent to the study and will directly address the research questions.

You rightly identify that we do not provide a lot of detail about the workshops with the purple/disability networks. We opted not to collect participant demographic or occupational information due to the highly sensitive, confidential and identifiable nature of the discussions in these workshops. Please see line 131 in the tracked changes document.

Reviewer 4 Report

Comments and Suggestions for Authors

***Please ensure that the abstract includes objective, methods, results, and conclusion. Population and sampling details are missing.

***There is no need for citation in the abstract section.

***Data were collected over a period of 7 months, from September 2021 to March 2022. Now is 2025; it is not up-to-date to explain the current situation. Please add the study's limitations.

***According to the methodology, please include the population and sampling technique. ***

***Each interview lasted approximately 60 minutes. Institutional Review Board Statement: The study was conducted in accordance with the Declaration of Helsinki and approved by the Departmental Ethics Committee of Work, Employment, and Organisation, Strathclyde Business School. Please add the approval serial number along with the approval date to ensure that the data was collected after the specified approval date.

***Please add some NVivo 10's results.

***One table is suggested to be added for demographics along with the date and time of data collection.

***The discussion should be done following 3.1-3.3 in thematic analysis regarding previous studies to support or argue.

***Please add the study's limitation regarding the data collection period in the heading, 5. Recommendations for Practice and Future Research.

Author Response

Reviewer 4

  1. Please ensure that the abstract includes objective, methods, results, and conclusion. Population and sampling details are missing.

We have amended the abstract.

  1. There is no need for citation in the abstract section.

We have amended the abstract.

  1. Data were collected over a period of 7 months, from September 2021 to March 2022. Now is 2025; it is not up-to-date to explain the current situation. Please add the study's limitations.

Thank you for this comment, we have added this to the limitations section. 

  1. According to the methodology, please include the population and sampling technique.

We have amended the methods section.

  1. Each interview lasted approximately 60 minutes. Institutional Review Board Statement: The study was conducted in accordance with the Declaration of Helsinki and approved by the Departmental Ethics Committee of Work, Employment, and Organisation, Strathclyde Business School. Please add the approval serial number along with the approval date to ensure that the data was collected after the specified approval date.

We have added an approval date for receiving favourable ethical opinion from the University of Strathclyde. The University does not generate serial numbers as part of its ethical review process.

  1. Please add some NVivo 10's results.

NVivo was used to facilitate the management of project data, it was not used to analyse the data, so there are no results to share beyond those presented in the paper.

  1. One table is suggested to be added for demographics along with the date and time of data collection.

We have added a table which details the years of experience of each of the participants as we believe this information to be relevant to the study. We do not share other demographic data other than to explain that the participants were observed to be predominately female. We have elected not to provide the specific times and dates of the interviews, only the time frame within which they were completed as is standard for peer review.

  1. The discussion should be done following 3.1-3.3 in thematic analysis regarding previous studies to support or argue.

We have made some revisions to our discussion section and added a conclusion section and section on research limitations.

  1. Please add the study's limitation regarding the data collection period in the heading, 5. Recommendations for Practice and Future Research.

We have added an additional limitations section.

Round 2

Reviewer 1 Report

Comments and Suggestions for Authors

ear Authors,

Thank you for submitting the revised version of your manuscript on supporting and retaining NHS staff with fluctuating long-term health conditions (FLTCs). I would like to commend you for the considerable improvements made in this version. The revisions have substantially strengthened the paper’s clarity, methodological transparency, and overall contribution.

Key Improvements

  • Sample and participant information: The inconsistencies noted in the earlier version have been resolved. The addition of detailed tables outlining participant roles, Trusts, and experience is especially helpful in clarifying the study’s scope.

  • Methodology and analysis: The abductive approach, coding process, and team validation are described in much more detail, with acknowledgement of the contested concept of “data saturation.” This enhances the study’s transparency and credibility.

  • Ethics and data protection: References have been updated to UK GDPR and the Data Protection Act 2018, ensuring accuracy and alignment with current standards.

  • Results structure: Findings are now clearly organised into three main themes, followed by a discussion of good practice. This makes the narrative more coherent and accessible.

  • Discussion and contribution: The revised version more clearly articulates the novel contribution of the study, particularly in showing how over-focus on condition trajectories may hinder effective support.

  • Limitations: The section is more candid and thorough, explicitly acknowledging the small sample, limited geography, lack of demographics, and absence of employee voices.

The manuscript now provides a clear, relevant, and timely contribution to the literature on HR management and workforce sustainability in the NHS. The improvements since the previous version are substantial, and only minor editorial and structural refinements remain. Thank you again for your thoughtful revisions and for contributing important insights into the management of fluctuating long-term health conditions in NHS England.

Author Response

The improvements since the previous version are substantial, and only minor editorial and structural refinements remain. 

Thank you for these comments. We are confident that the minor editorial and structural refinements will be addressed as we finalise the document for submission with the journal team.

Reviewer 4 Report

Comments and Suggestions for Authors

***Reviewer Comment 6: Please add some NVivo 10's results.

Author's Response 6: 

NVivo was used to facilitate the management of project data, it was not used to analyse the data, so there are no results to share beyond those presented in the paper.

Reviewer Comment to Author's Response 6: 

If there is no NVivo's results, it is suggested to removed NVivo out of the methodology.

***Reviewer Comment 7: 

One table is suggested to be added for demographics along with the date and time of data collection.

Author's Response 7: 

We have added a table which details the years of experience of each of the participants as we believe this information to be relevant to the study. We do not share other demographic data other than to explain that the participants were observed to be predominately female. We have elected not to provide the specific times and dates of the interviews, only the time frame within which they were completed as is standard for peer review.

Reviewer Comment to Author's Response 7: 

Please add this statement in the manuscript.

The authors do not share other demographic data other than to explain that the participants were observed to be predominately female. The methodology process has elected not to provide the specific times and dates of the interviews, only the time frame within which they were completed, as is standard for peer review.

***Other comments are addressed.***

Author Response

    1. (Please add some NVivo 10's results.)

    If there is no NVivo results, it is suggested to remove NVivo out of the methodology.

    Thank you for this comment. Though we did not use NVivo to analyse our data, we did use it to manage our data. We have clarified this in the paper. Please see lines 212-216 in the tracked changes document.

    1. (One table is suggested to be added for demographics along with the date and time of data collection.)

    Please add this statement in the manuscript. “The authors do not share other demographic data other than to explain that the participants were observed to be predominately female. The methodology process has elected not to provide the specific times and dates of the interviews, only the time frame within which they were completed, as is standard for peer review.”

    Thank you. We have added this to the paper. Please see lines 148-159 in the tracked changes document.